# Modeling and Design of the Automatic Pressure Regulating Valve in the Foam Firefighting System

**Zhijun Yang [1], Hongjun Chen [2], Lifang He [1] and Xikun Wang [1],***

[1] Research Center of Fluid Machinery Engineering and Technology, Jiangsu University, Zhenjiang 212013, China; yangzhijun@shfri.cn (Z.Y.); helifang222@163.com (L.H.)
[2] Chongqing Pump Industry Co. Ltd., Chongqing 400033, China; chenhj@cqpump.com
* Correspondence: wangxk@ujs.edu.cn

**Abstract:** This paper presents a novel design of the automatic pressure balancing valve, used in the in-line balanced pressure (ILBP) proportioner for the foam firefighting system, at a required percentage of solution. Featured in a four-chamber configuration with a double-acting diaphragm actuator, it can automatically maintain the foam concentrate pressure with the pressure in the supply water pipeline, within a precision level of 0.02 MPa (or 1.3%), under the design operating condition. The static characteristics at the equilibrium state have been discussed in terms of poppet displacement with reference to the geometrical dimensions and operating pressures of the valve. The dynamic response of the valve during the startup has been examined through building the mathematical model of the forces on the valve and solving it numerically using MATLAB. The results show that the response time of the valve is always less than 0.01 s, which fully satisfies the stability and hysteresis requirement. The prototype has been tested in the laboratory, which agrees well with the numerical results. It was then successfully put into production, forming the first series of ILBP foam pump firefighting system in China.

**Keywords:** automatic pressure balancing valve; foam firefighting system; in-line balanced pressure (ILBP) proportioner; mathematical model; simulation

---

## 1. Introduction

Protein-based firefighting foams provide a robust foam blanket for Class B fire and vapor suppression. The in-line balanced pressure (ILBP) proportioner for the foam firefighting system is widely used in areas of flammable and combustible liquids, which are bulk stored, processed, or handled [1]. A complete ILBP foam firefighting system is shown in Figure 1, which consists of the ILBP proportioner, an atmospheric foam concentrate storage tank(s), positive foam concentrate pump(s), pump controller(s) and valves, piping, etc. The ILBP proportioning system uses a positive pressure foam injection, requiring the foam concentrate to be supplied by the foam pump at a higher pressure ($P_0$) than the water supply ($P_1$), i.e., $P_0 > P_1$. During real applications, however, $P_1$ may vary, thus requiring a valve to automatically adjust the foam concentrate pressure. As the key device in the ILBP proportioner, the role of the automatic pressure balancing valve is to automatically adjust the incoming foam concentrate pressure ($P_0$) to an output value ($P_2$) that keeps consistent with $P_1$, before entering the foam proportioning mixer to produce a foam solution with a proper concentration to extinguish fire efficiently.

This type of firefighting system with a foam concentrate pump has several advantages, including: (1) The foam concentrate can be pumped longer distances with the pressure balancing valve being kept adjacent to the proportioner; and (2) the foam supply can be replenished on-the-run by adding to the atmospheric tank [1]. There are several ILBP system suppliers, such as Chemguard [2] and

Ansul [3], however, there is no ILBP manufacturer in China, while there are increasing concerns about risks related with flammable liquids (e.g., GB 30000.7-2013) [4]. The ILBP proportioner, together with other devices in the firefighting system, should be designed properly according to the requirement of specific applications (including flow capacity, output pressure, response time, percentage of foam water solution, etc.), which may vary case by case. Among these devices, the pressure balancing valves with high accuracy and reliability are crucial. This paper presents our design of a pressure balancing valve catered for a relatively large capacity (up to 12 L/s foam concentrate or 200 L/s foam solution) and high operating pressure ($P_0$ = 1.76 MPa, $P_1$ = 1.6 MPa).

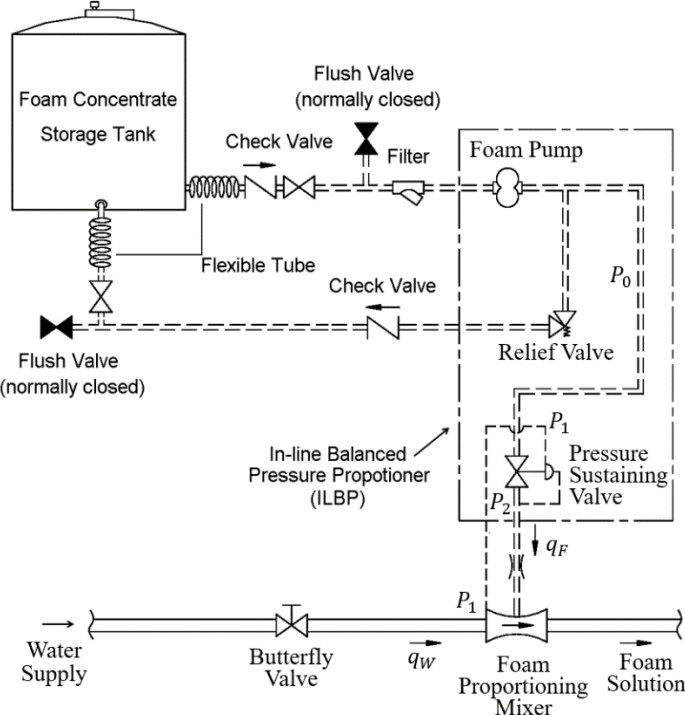

**Figure 1.** Schematic of the in-line balanced pressure (ILBP) proportioning system.

Two of the most common designs for pressure control valves are the poppet- and spool-type. The present study adopts the poppet type due to the following considerations. Poppet valves, as compared to spool valves, require less stringent machining tolerances, are less susceptible to contamination problems, have very low leakage, which make it possible to eliminate two separate supply lines [5] and hence, have been widely used for pressure relief [6,7]. It is natural to design the valve as the diaphragm-spring-poppet configuration as shown in Figure 2a, which is flow-to-open, spring-to-close. It consists of two distinct parts: The (lower) body with springs to provide the restoring force; and the (upper) double-acting actuator, in which the upward/downward motion of the diaphragm is controlled by the hydraulic pressure difference in the two chambers connecting to the water supply ($P_1$) and the valve outlet ($P_2$), respectively. Following the guiding techniques for the design of poppet valves [8], a prototype based on this design has been manufactured and tested in the laboratory. Unfortunately, the valve could not satisfy the requirement in that the difference between $P_1$ and $P_2$ exceeds the allowable limit. This is attributed to the long-standing drawback of spring-loaded poppet valves—the instability issue [9]. Researchers have conducted many simulations and experimental studies to investigate the underlying mechanism of instability and its consequence. Funk [10] claimed that instability is caused by an interaction between the poppet spring-mass system and line dynamics. Hayashi [9] suggested that it is important to examine valve motions with a small valve lift and hysteresis of flow forces, while some researchers (e.g., Han et al. [11]) reported that cavitation also plays a role. In more recent years, there is a trend of using the computational fluid dynamics (CFD) tool

to simulate the dynamic behavior of poppet valves (e.g., Gomez et al. [12]) and using the experimental visualization method to directly capture the valve vibrations [13,14]. Instability is an inherent issue for the spring-loaded poppet valves, which cannot be eliminated completely. Therefore, a novel design of the pressure balancing valve is proposed, as shown schematically in Figure 2b. In this design, the spring has been completely removed. Rather, an additional chamber, called the cushion chamber, is created above the poppet (tapered plug). Hence, it is featured in a four-chamber configuration with a double-acting diaphragm actuator. Details about its working principle and design considerations will be discussed in the next section.

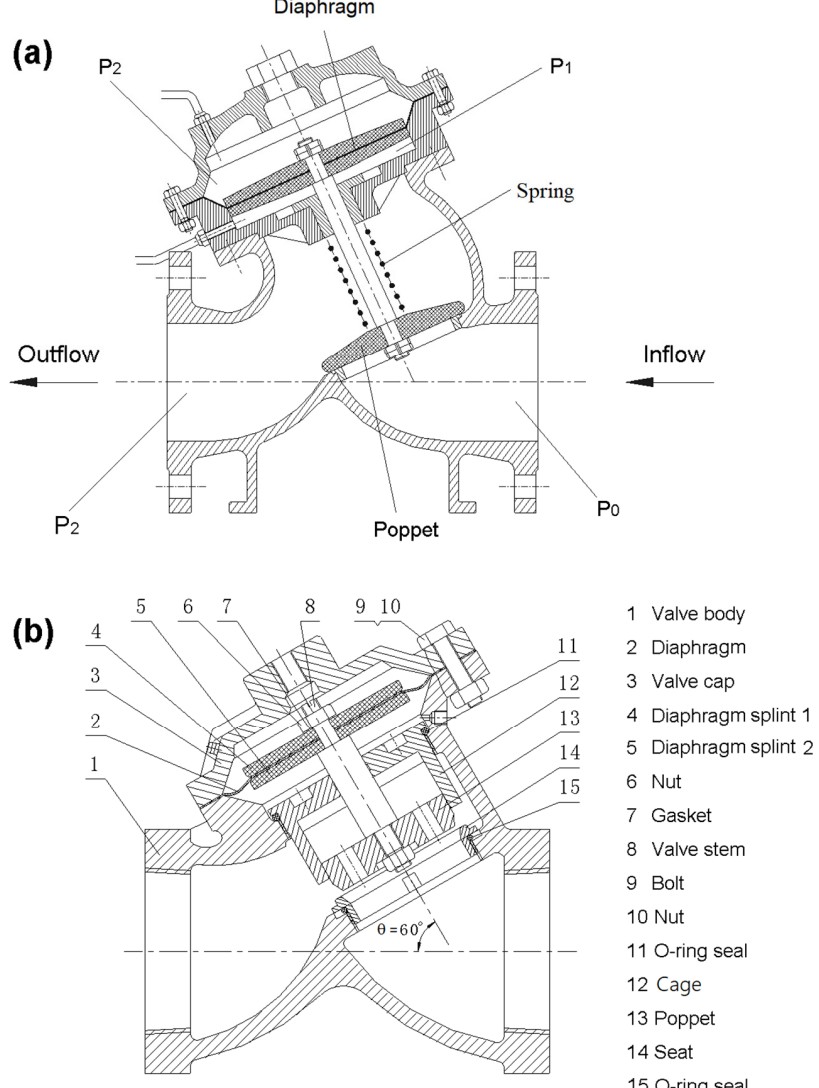

**Figure 2.** Design of the pressure-sustaining diaphragm valve: (**a**) Conventional type; (**b**) novel type.

## 2. Mathematical Model of Valve Motion

Forces acting on the proposed four-chamber poppet valve are provided in Figure 3. The differential pressure force on the two sides of the diaphragm drives the motion of the valve (stem and poppet) along the valve axis ($X$-direction). Originally, the poppet remains on the seat in the closed position. When there is a sudden supply of water in the main pipeline (with pressure $P_1$), the diaphragm will be pushed upward, and thus the poppet is lifted off from the seat with a displacement ($X$), forming an orifice (which is the valve opening in the ring shape) through which the pressurized foam concentrate (with pressure $P_0$) passes from the supply chamber to the outlet port (with pressure $P_2$). Meanwhile,

as the poppet moves upward, the fluid trapped in the cushion chamber is squeezed and the pressure inside ($P_X$) begins to build up, generating the reaction force to deaccelerate the motion of the poppet. Note that both $P_2$ and $P_X$ are a function of time. In order to avoid a rather abrupt change in $P_2$ or $P_X$, there are four diversion holes (with diameter $d_2$) through the poppet and a narrow gap between the poppet and the cage, which connects the cushion chamber with the supply chamber and with the outlet port, respectively. The interlinkage of fluid domains serves as a buffer for the valve motion, leading to a more gradual yet fast enough response until the steady state (equilibrium) is reached.

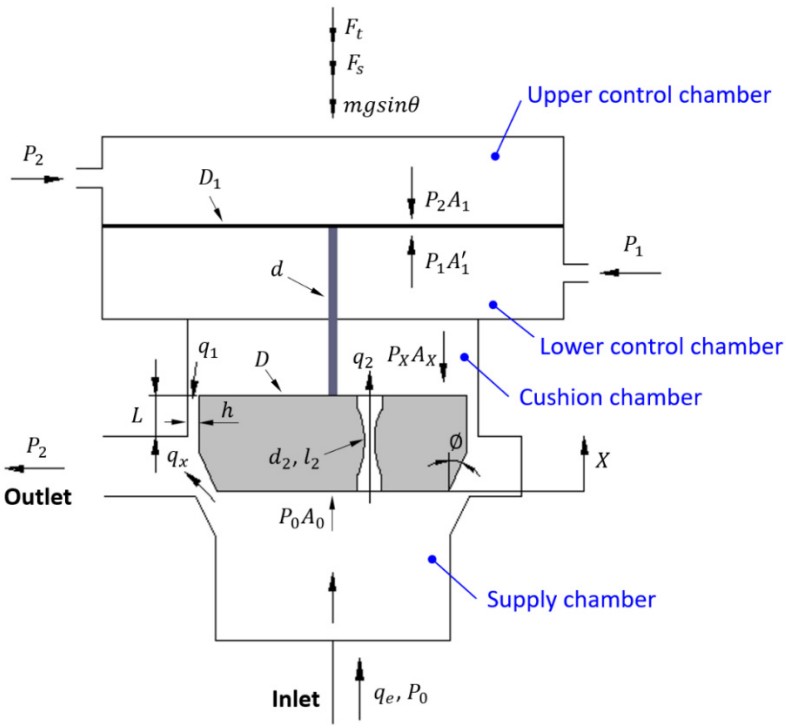

**Figure 3.** Schematic of forces on the pressure-sustaining diaphragm valve.

According to Figure 3, the static forces acting on the valve when reaching the equilibrium satisfy:

$$P_0 A_0 + P_1 A_1' = P_X A_X + P_2 A_1 + mgsin\theta + F_s \tag{1}$$

where $A_0$, $A_X$, $A_1$, and $A_1'$ are the cross-sectional areas of two sides of the poppet and diaphragm subjective to the fluid pressure; $m$ is the mass of the valve; $g$ is the gravitational acceleration; $\theta$ (=60°) is the inclined angle of the valve axis with respect to the horizontal direction; and $F_s$ is the steady-hydrodynamic fluid force acting on the poppet, which is a function of the poppet geometry, pressure drop through the valve opening $\Delta P$ ($= P_0 - P_2$), and the poppet displacement ($X$).

The re-arrangement of Equation (1) leads to the equations for the outlet pressure ($P_2$) and the difference between $P_2$ and $P_1$:

$$P_2 = \frac{A_1'}{A_1} P_1 + \frac{P_0 A_0 - P_X A_X}{A_1} - \frac{mgsin\theta + F_S}{A_1} \tag{2}$$

$$\Delta P_{1,2} = \frac{A_1' - A_1}{A_1} P_1 + \frac{P_0 A_0 - P_X A_X}{A_1} - \frac{mgsin\theta + F_S}{A_1} \tag{3}$$

It can be seen from Equation (3) that with the other parameters kept constant, a higher value of $A_1$ (or larger diaphragm) is beneficial to reduce $\Delta P_{1,2}$. Since $A_1' = A_1 - \pi d^2$ (where $d$ is the diameter of the valve stem), $A_1' < A_1$ holds so that the first term on the right-hand side of Equation (3) is always negative. The same is with the third term, whereas the only positive term is the second term.

Therefore, it is required to carefully choose the structural parameters to ensure that the arithmetic sum of the three terms approaches zero [15]. For instance, in order to decrease the magnitude of the first term, it is desirable to reduce the diameter of the valve stem $d$. On the other hand, if $d$ is too small, the stem would be too weak to withstand the hydraulic pressure force that can be as high as the order $\sim 10^4$ N. The design of the valve needs to be considered comprehensively from both the functional and structural perspectives.

In addition to the static characteristics, an even more important consideration for assessing the performance of the valve is its dynamic characteristics [5], including the fluctuation range of the output pressure ($P_2$) and the time taken to reach equilibrium (called response time or time lag). The dynamic characteristics are affected by many factors, including the geometric parameters of the valve and the operating conditions. Physical modeling with hardware and prototypes can be effective, but it is very time consuming and costly. The model-based design is an alternative to creating physical prototypes for evaluating the dynamics of the proposed system. Several rounds of numerical optimization had been conducted before building the physical prototype, which was tested in the laboratory as shown in Figure 4. The main structure parameters of the prototype are provided in Table 1.

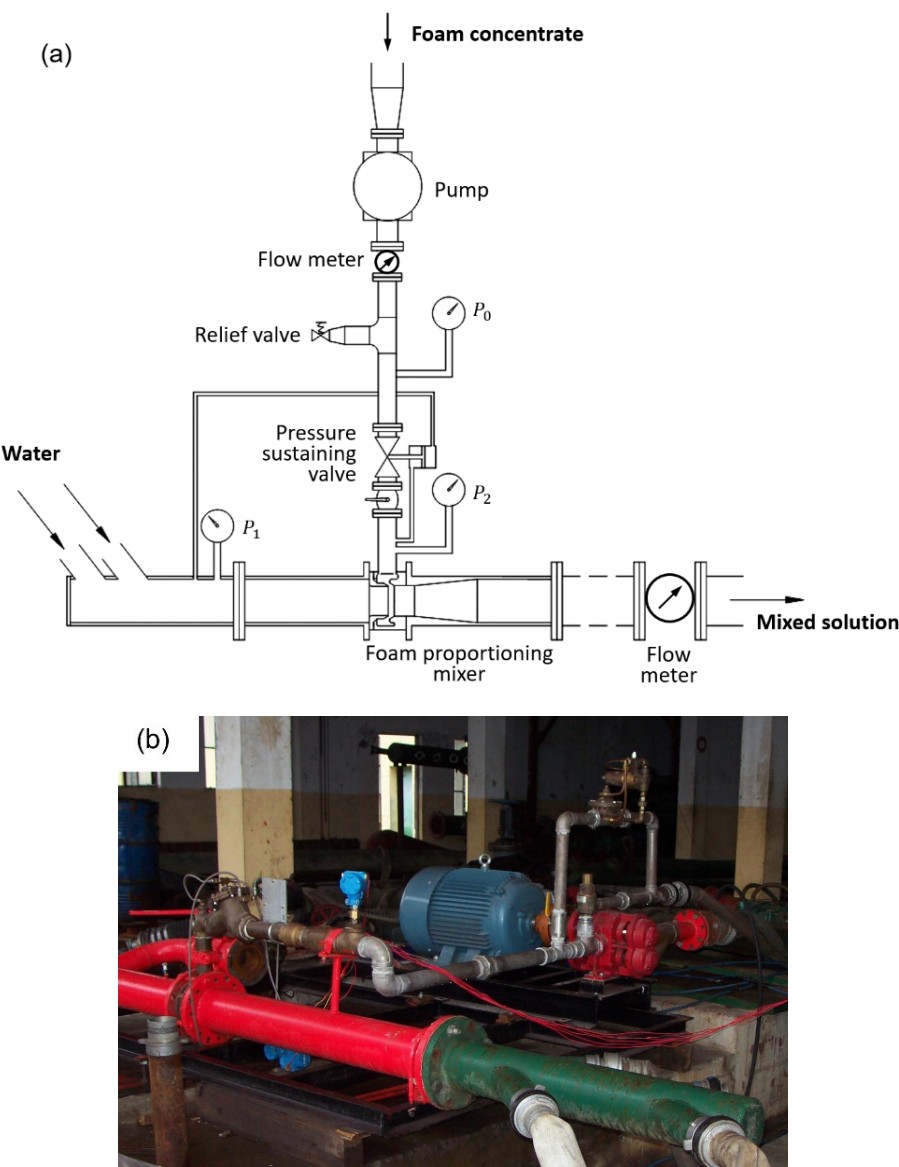

**Figure 4.** Prototype test: (**a**) Schematic of the test setup; (**b**) laboratory test rig.

**Table 1.** Main structure dimensions of the valve.

| Symbol | Physical Quantity | Values Selected |
|---|---|---|
| $A_0$ | Flow area of the valve seat ($m^2$) | $2.9 \times 10^{-3}$ |
| $A_X$ | Sectional area of the upper side of the poppet ($m^2$) | $2.8 \times 10^{-3}$ |
| $A_1$ | Sectional area of the upper side of the diaphragm ($m^2$) | $9.6 \times 10^{-3}$ |
| $A_1'$ | Sectional area of the lower side of the diaphragm ($m^2$) | $9.4 \times 10^{-3}$ |
| $A_2$ | Sectional area of the diversion hole ($m^2$) | $5 \times 10^{-5}$ |
| $D$ | Diameter of the poppet (m) | 0.064 |
| $D_1$ | Diameter of the diaphragm (m) | 0.11 |
| $d$ | Diameter of the stem (m) | 0.016 |
| $d_2$ | Diameter of the diversion hole (m) | 0.008 |
| $H$ | Height of the cushion chamber (m) | 0.02 |
| $h$ | Thickness of the gap between the poppet and the cage (m) | 0.00005 |
| $L$ | Length of the gap between the poppet and the cage (m) | 0.01 |
| $V_X$ | Volume of the cushion chamber ($m^2$) | $6 \times 10^{-5}$ |
| $\varnothing$ | Half angle of the poppet (°) | 30 |
| $\theta$ | Incline angle of the valve axis (°) | 60 |

Dynamic models of the pressure balancing valves have added complexity due to their non-linear nature, inherent hysteresis, and compressibility of the fluid media. This added complexity makes existing predefined modeling software packages less than ideal for developing hydraulic system simulations. In order to simplify the analysis, the following assumptions are made: (i) The working fluid is water with constant density ($\rho$) and bulk modulus (K); (ii) the gap width between the poppet and the cage is uniform, since the non-uniform gap would induce the lateral force and hence, lateral vibration of the poppet [16]. The following are the mathematical models built for this valve:

(1) Flow equation through the valve opening:

$$q_X = \alpha_{DX} \pi D X sin \varnothing \sqrt{2 \Delta P / \rho} = b_1 \sqrt{\Delta P} X \tag{4}$$

where $q_X$ is the flow rate, $\Delta P$ ($= P_0 - P_2$) is the pressure difference between the supply chamber and the outlet port, and $\alpha_{DX}$ (=0.6) is the discharge coefficient. Hence, we can calculate the lumped coefficient as $b_1 = \alpha_{DX} \pi D X sin \varnothing \sqrt{2/\rho}$.

(2) Flow discharge equation for the four diversion holes through the poppet:

$$q_2 = 4C_{d2}A_2 \sqrt{\frac{2(P_0 - P_X)}{\rho}} = 4b_3 \sqrt{P_0 - P_X} \tag{5}$$

where $A_2$ ($= \frac{\pi d_2^2}{4}$) is the cross-sectional area of the diversion hole, $C_{d2}$ (=0.65) is the discharge coefficient, and $b_3$ ($= C_{d2}A_2 \sqrt{2/\rho}$) is the lumped coefficient.

(3) Flow equation for leakage through the gap between the poppet and the cage:

$$q_1 = K_v (P_X - P_2) \tag{6}$$

where $K_v$ is the leakage flux coefficient, $K_v = \frac{\pi D (2h)^3}{96 \mu L} + \pi D h u = C_1 + C_2 \frac{dX}{dt}$, and $C_1$ and $C_2$ are constants.

(4) Continuity equation for flow in the supply chamber:

$$q_e = q_X + q_2 + A_0 \frac{dX}{dt} + \frac{V_0}{K} \frac{dP_0}{dt} \tag{7}$$

where $V_0$ is the volume of fluid in the supply chamber. Since the inlet pressure $P_0$ is kept constant via the relief valve, $\frac{dP_0}{dt}$ is considered to be zero. Then, we can get:

$$q_X = q_e - q_2 - A_0 \frac{dX}{dt} \tag{8}$$

(5) Continuity equation for flow in the cushion chamber:

$$q_2 = q_1 - A_X \frac{dX}{dt} + \frac{V_X}{K} \frac{dP_X}{dt} \tag{9}$$

where $V_X$ is the volume of fluid in the cushion chamber, $V_X = \frac{\pi(D^2 - d^2)(H - X)}{4} = C_3 + C_4 X$, and $C_3$ and $C_4$ are constants.

(6) Ordinary differential equation (ODE) for valve motion:

$$P_0 A_0 + P_1 A_1' - P_X A_X - P_2 A_1 = m \frac{d^2 X}{dt^2} + B \frac{dX}{dt} + mg\sin\theta + F_t + F_s \tag{10}$$

where $B$ is the damping coefficient due to the viscosity of fluid in the gap; $F_t$ and $F_s$ are the instantaneous and steady fluid forces acting on the poppet, respectively. These quantities can be calculated as $B = \frac{\mu A}{h} = \frac{2\pi\mu DL}{h}$, in which $\mu$ is the dynamic viscosity of fluid and $A$ is the area of contact between the poppet and the cage; $F_t = \rho L d \frac{dq_X}{dt} \approx \rho L b_1 \sqrt{\Delta p} \frac{dX}{dt}$; $F_s = \alpha_{DX} C_{vX} \pi DX \sin 2\varnothing \Delta p = b_2 \Delta PX$, in which $C_{vX}$ (=0.98~0.99) is the flow velocity coefficient and $b_2 = \alpha_{DX} C_{vX} \pi D \sin 2\varnothing$.

Based on the above equations, we can obtain expressions for the following parameters: $P_2$, $P_X$, and $X$. Based on Equation (4), $\Delta P = \left(\frac{q_X}{b_1 X}\right)^2$ where $q_X$ can be obtained from Equation (8), and then we can get:

$$P_2 = P_0 - \Delta P = P_0 - \left(\frac{q_e - 4b_3\sqrt{P_0 - P_X} - A_0\frac{dX}{dt}}{b_1 X}\right)^2 \tag{11}$$

The re-arrangement of Equations (9) and (10) leads to:

$$\frac{dP_X}{dt} = \frac{K}{C_3 - C_4 X}\left[A_X \frac{dX}{dt} - \left(C_1 + C_2 \frac{dX}{dt}\right)(P_X - P_2) + 4b_3\sqrt{P_0 - P_X}\right] \tag{12}$$

$$\frac{d^2 X}{dt^2} = \frac{1}{m}\left[P_0 A_0 + P_1 A_1' - P_X A_X - P_2 A_1 - mg\sin\theta - B\frac{dX}{dt} - \rho L b_1 \sqrt{P_0 - P_2}\frac{dX}{dt} - b_2(P_0 - P_2)X\right] \tag{13}$$

Equations (11)–(13) are the mathematical model for the dynamics of the valve, which is a system of 2nd-order non-linear ODEs. A MATLAB code using the ode45 solver has been written to solve these equations. During the numerical simulations, the step of integration was 0.0005 s and the root-mean-square (RMS) residual level of $10^{-4}$ was set as the convergence criteria.

## 3. Results and Analysis

It is noted that some of the coefficients in these equations depend not only on the valve geometry, but also on the fluid medium. Therefore, two different types of working fluid, i.e., water and protein foam concentrate, have been considered. The two types of fluid have similar compressibility (with bulk modulus $K = 2.2 \times 10^9$ Pa/m$^2$), but the protein foam concentrate has a relatively higher density and viscosity than water (i.e., density $\rho = 1160$ vs. 1000 kg/m$^3$ and dynamic viscosity $\mu = 6 \times 10^{-3}$ vs. $1 \times 10^{-3}$ Pa·s). Figure 5 presents the dynamic behavior (including poppet displacement $X$ and outlet pressure $P_2$) during the start-up under the design condition (flow rate $q_e = 8$ L/s and control pressure $P_1 = 1.6$ MPa). For these simulations, the inlet pressure is kept constant at $P_0 = 1.76$ MPa and the initial outlet pressure $P_2(0) = 0$ Pa. The valve is assumed to be opening from its rest position, so the initial values of (0, 0) is used for ($X$, $dX/dt$). It can be seen that the valve opens quickly within 0.01 s,

after which both the outlet pressure ($P_2$) and the poppet displacement ($X$) approach the equilibrium values (denoted as $\widetilde{P_2}$ and $\widetilde{X}$). In other words, the response time of the valve is less than 0.01 s. At the beginning of valve opening, $P_2$ increases sharply forming a spike (or an overshoot) and then levels off at $\widetilde{P_2}$ = 1.61 MPa, which is very near the target value of 1.6 MPa. The variation in the fluid medium, water or protein foam concentrate, seems to barely affect the outlet pressure, since the two curves almost coincide with each other. However, it does affect the value of equilibrium poppet displacement, which is $\widetilde{X}$ = 5.7 and 6.3 mm for the water and protein foam concentrate, respectively. Considering that the protein foam concentrate is used in real applications, the following simulations select the protein foam concentrate as the working fluid.

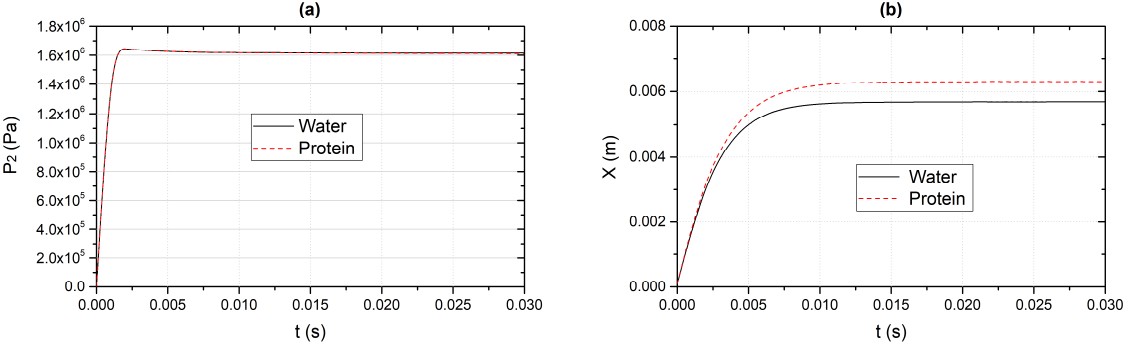

**Figure 5.** Comparison of the start-up characteristics due to the change in fluid medium—water and protein: (**a**) Outlet pressure $P_2$; (**b**) poppet displacement $X$. For these simulations: $q_e$ = 8 L/s and $P_1$ = 1.6 MPa.

Figure 6 presents the valve's start-up characteristics under different operating conditions, $P_1$ = 1.2 and 1.6 MPa and $q_e$ = 5 and 10 L/s. It can be seen that the dynamic characteristics are similar to that in Figure 5 and the response time maintains constant at 0.01 s. The value of $\widetilde{P_2}$ seems to be unaffected by the flow rate, but slightly varies with the control pressure, namely, $\widetilde{P_2}$ ≈ 1.28 and 1.61 MPa for $P_1$ = 1.2 and 1.6 MPa, respectively. On the other hand, the value of equilibrium poppet displacement $\widetilde{X}$ increases with both $q_e$ and $P_1$. The effects of varying initial conditions (ICs) on the dynamical response of the valve are examined in Figure 7, with $P_2(0)$ = 0, 1.5, and 1.7 MPa and $X(0)$ = 0, 3, and 5 mm. Regardless of the ICs, both $P_2$ and $X$ quickly return to the equilibrium within the response time of about 0.01 s, either without or with one oscillation, indicating that the system is overdamped with good stability.

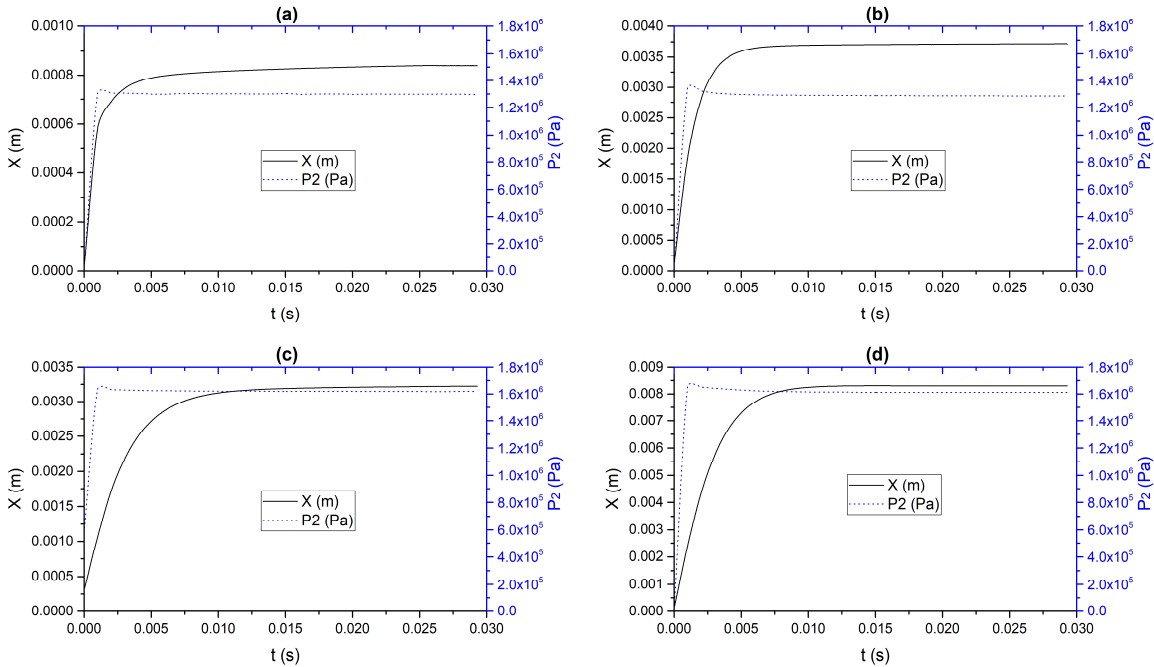

**Figure 6.** Development of poppet displacement $X$ (left-side $y$-axis) and outlet pressure $P_2$ (right-side $y$-axis) during the start-up for: (**a**) $q_e$ = 5 L/s and $P_1$ = 1.2 MPa; (**b**) $q_e$ = 10 L/s and $P_1$ = 1.2 MPa; (**c**) $q_e$ = 5 L/s and $P_1$ = 1.6 MPa; (**d**) $q_e$ = 10 L/s and $P_1$ = 1.6 MPa.

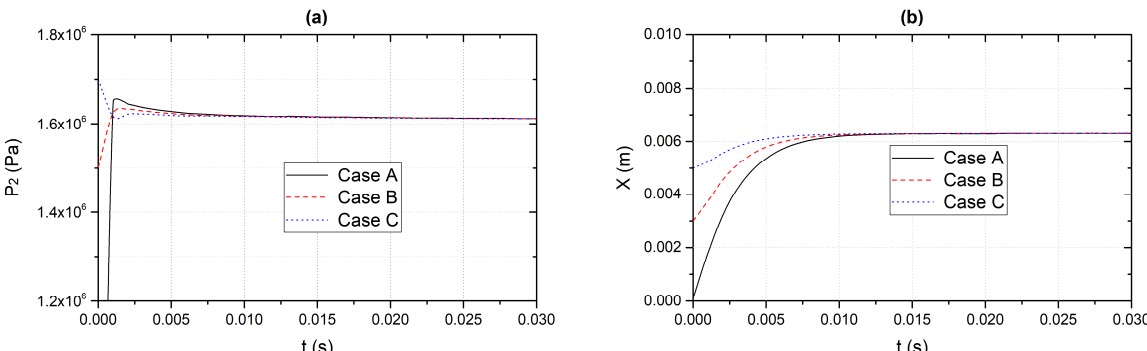

**Figure 7.** Effect of varying initial conditions (ICs) on the valve start-up characteristics under $q_e$ = 8 L/s and $P_1$ = 1.6 MPa: (**a**) Outlet pressure $P_2$; (**b**) poppet displacement $X$. ICs for Case A: $P_2(0)$ = 0 Pa and $X(0)$ = 0 mm; ICs for Case B: $P_2(0)$ = 1.5 MPa and $X(0)$ = 3 mm; ICs for Case C: $P_2(0)$ = 1.7 MPa and $X(0)$ = 5 mm.

The present study also investigates the effects of varying geometrical parameters on the valve's static and dynamic properties. The first parameter is the diameter of the diversion holes $d_2$. The four diversion holes connecting the supply chamber and the cushion chamber are introducing some pressure drop between the two chambers, while the small clearance between the poppet body and the cage introduces a high viscous friction force to increase the stability of the valve. When the valve reaches equilibrium, the following relationships hold: $P_0 > P_X > P_2$ and $q_1 = q_2$. The latter can be written as the following equation by setting the time derivative term in Equation (6) to zero:

$$4b_3 \sqrt{P_0 - P_X} = C_1(P_X - P_2) \tag{14}$$

where $b_3 = C_{d2}\frac{\pi d_2^2}{4}\sqrt{2/\rho}$ and $C_1 = \frac{\pi D(2h)^3}{96\mu L}$.

There exists a minimum value of $b_3$ (or $d_2$) to ensure the valid solution of $P_X \in (P_2, P_0)$ satisfying Equation (14). Figure 8a indicates that the critical diameter of diversion holes is found to be 7~7.5 mm;

at $d_2 \geq 7.5$ mm, $P_2$ is very stable with a constant equilibrium value of $\widetilde{P_2} \approx 1.61$ MPa, which is almost unaffected by the change in $d_2$; at $d_2 = 7$ mm, however, $P_2$ experiences a sudden decrease at $t = 0.015$ s, attaining a lower equilibrium value of $\widetilde{P_2} = 1.59$ MPa. When the diameter of the diversion holes is further decreased to $d_2 \leq 6$ mm, $\widetilde{P_X}, \widetilde{P_2}$ and $\widetilde{X}$ become complex numbers, suggesting that the diversion holes are too small to ensure physically meaningful results. On the other hand, the increase of $d_2$ results in a monotonic decrease of $\widetilde{X}$, as shown in Figure 8b. A higher value of $\widetilde{X}$, as long as the maximum value does not exceed the allowable limit of poppet displacement (which is 10 mm), is desirable to increase the sensitivity of the valve. Therefore, $d_2$ is set at 8 mm.

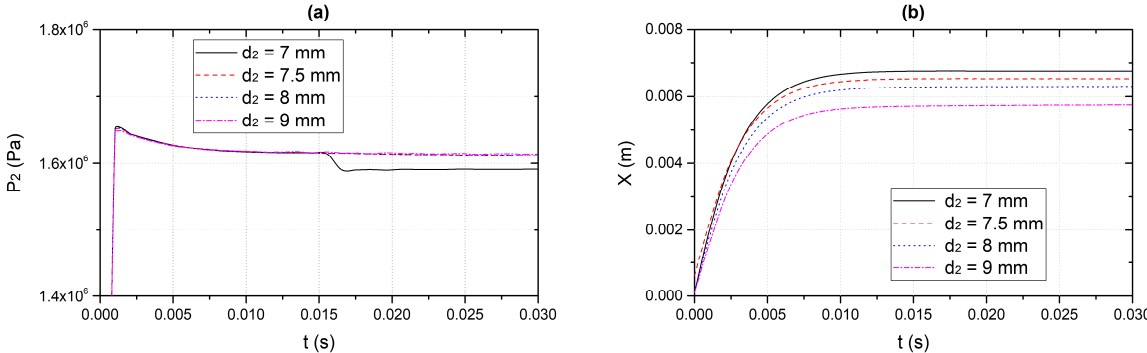

**Figure 8.** Effect of varying the diameter of the diversion holes ($d_2 = 7, 8, 9$, and 10 mm) on the valve start-up characteristics under $q_e = 8$ L/s and $P_1 = 1.6$ MPa: (**a**) Outlet pressure $P_2$; (**b**) poppet displacement $X$.

Figure 9 presents the effects of varying the diameter of valve stem $d$ (= 8, 16, and 24 mm). As opposed to the initial expectation that the slender stem might be beneficial to minimize the difference between $A_1'$ and $A_1$, and hence the pressure difference between $P_2$ and $P_1$, the diameter of the valve stem nearly has no effect on the responsiveness of the valve. The reason is that $d$ does not only affect $A_1'$, but also affects $A_X$, $V_X$, and $m$. Finally, $d = 16$ mm is selected based on the material strength requirement under the highest pressure.

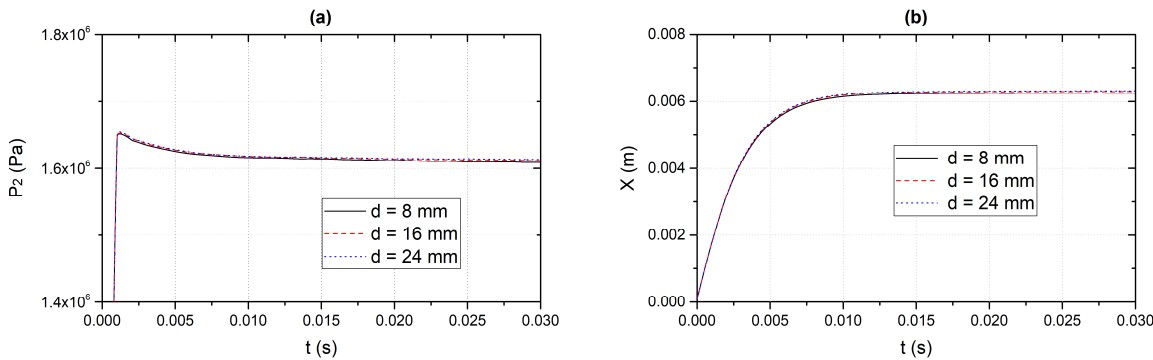

**Figure 9.** Effect of varying the diameter of the valve stem ($d = 8, 16$, and 24 mm) on the valve start-up characteristics under $q_e = 8$ L/s and $P_1 = 1.6$ MPa: (**a**) Outlet pressure $P_2$; (**b**) poppet displacement $X$.

Figure 10 shows the variation of the equilibrium outlet pressure $\widetilde{P_2}$ and poppet displacement $\widetilde{X}$ versus the inlet flow rate ($q_e$) when the control pressure $P_1$ is fixed at 1.6 MPa. It is shown that $\widetilde{P_2}$ slightly decreases with $q_e$, from $\widetilde{P_2} \approx 1.62$ MPa at $q_e = 4$ L/s to $\widetilde{P_2} \approx 1.605$ MPa at $q_e = 12$ L/s. It fully satisfies the design requirement that the maximum difference between $\widetilde{P_2}$ and $P_1$, or adjustment error $e_P = \widetilde{P_2} - P_1$, is ±0.1 MPa. In practice, $\widetilde{P_2}$ is desirable to be slightly higher than $P_1$, since the output foam concentrate from the valve would experience some pressure drop before entering the proportioner due to a viscous loss in the pipeline. The equilibrium poppet displacement $\widetilde{X}$ increases monotonically with $q_e$, from about 2 mm at $q_e = 4$ L/s to 10 mm at $q_e = 12$ L/s. Moreover, plotted in this figure is the theoretical

value based on the static analysis, which can be obtained by plugging $F_s = b_2 \Delta PX = b_2\left(P_0 - \widetilde{P_2}\right)\widetilde{X}$ into Equation (1):

$$\widetilde{X} = \frac{1}{b_2\left(P_0 - \widetilde{P_2}\right)}\left(P_0 A_0 + P_1 A_1' - \widetilde{P_X}A_X - \widetilde{P_2}A_1 - mgsin\theta\right) \tag{15}$$

It is noted that there is a small difference between the numerical and theoretical results, which is believed to be due to round-off errors and truncation errors during computations.

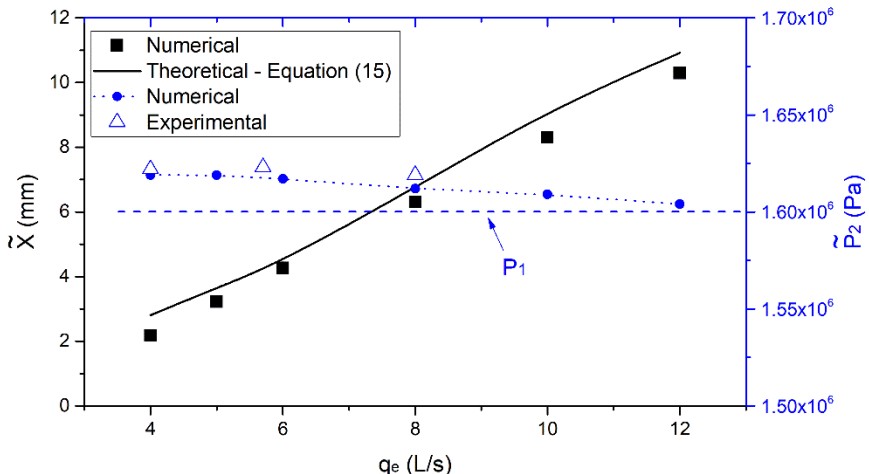

**Figure 10.** Variation of the equilibrium poppet displacement $\widetilde{X}$ and outlet pressure $\widetilde{P_2}$ versus the inlet flow rate ($q_e$) for $P_1 = 1.6$ MPa.

Figure 11 shows that when the control pressure $P_1$ is 1.2 MPa, $\widetilde{P_2} \approx 1.3 \sim 1.28$ MPa which also slightly decreases with the flow rate. As compared to the case of $P_1 = 1.6$ MPa, the pressure adjustment error is relatively higher, i.e., $e_P \approx 0.08 \sim 0.1$ MPa, but still within the allowable limit. The dynamic responses of the valve have been tested on the prototype, for which the protein foam concentrate was used as the working fluid. Figure 12a presents the measured pressure signals ($P_0$, $P_1$, and $P_2$) for the case of varying $P_1$ from 1.2 to 1.6 MPa under the fixed flow rate ($q_e = 8$ L/s) and supply pressure ($P_0 = 1.76$ MPa). At $0 \le t \le 6$ s or $12 \le t \le 18$ s where $\widetilde{P_1}$ is fixed at about 1.2 and 1.6 MPa, respectively, $P_2$ also maintains a constant level which is slightly higher than $P_1$. More importantly, at $6 \le t \le 12$ s where $P_1$ significantly changes, either continuously or abruptly, $P_2$ always follows $P_1$ without a noticeable time lag. According to the laboratory tests under a wide array of operating conditions, the magnitude of $e_P$ is always less than 0.05 MPa, completely satisfying the design requirement. The effects of varying flow parameters (set pressures, flow rate, etc.) on the dynamic instability characteristic of the valve have been evaluated similar to the procedure described by Ma et al. [17]. Then, this valve prototype was successfully put into production and assembled with other components, forming the first series of commercial ILBP proportioning unit in China, as shown in Figure 12b. The slight over-prediction of $\widetilde{P_2}$ by the numerical simulation might arise from the relatively higher viscosity of the fluid medium used. Note that the discharge equations such as Equations (4)–(6), as well as the coefficients in those equations, were derived based on previous measurement data for water or air, the viscosity of which is negligibly small. Nevertheless, the flow is always tied to the fluid viscosity, the higher the fluid viscosity, the smaller the flow rate through orifices or pipes at the same pressure difference. For example, Nguyen et al. [18] found that the flow coefficient of a valve depends not only on the valve geometry and valve opening but also on the Reynolds number. Therefore, the simulation of the flow field inside the valve using CFD tools will be conducted to illustrate the viscous effect due to the variation in the working fluid, as well as the cavitation characteristics, which are important in hydraulic poppet valves [11,19,20].

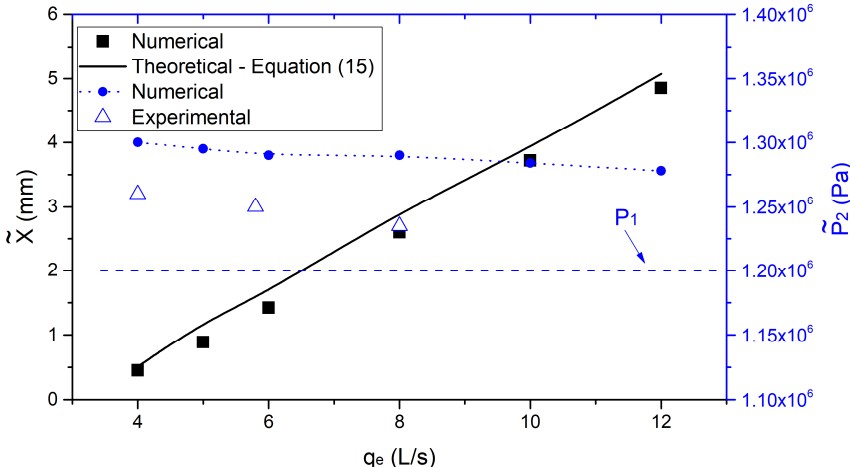

**Figure 11.** Variation of the equilibrium poppet displacement $\widetilde{X}$ and outlet pressure $\widetilde{P_2}$ versus the inlet flow rate ($q_e$) for $P_1$ = 1.2 MPa.

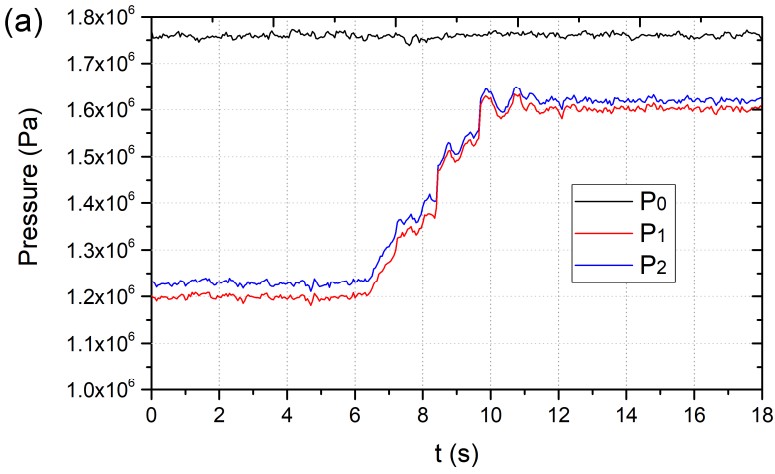

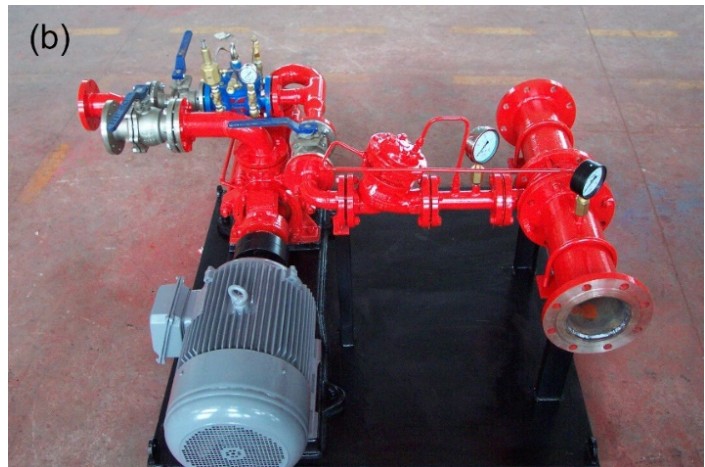

**Figure 12.** (**a**) Time history of measured pressure signals ($P_0$, $P_1$, and $P_2$) with $q_e$ fixed at 8 L/s and $P_1$ varied from 1.2 to 1.6 MPa; (**b**) the developed IBPP unit—BPPH6/150 series.

## 4. Conclusions

This paper presents our design of the automatic pressure balancing valve used in the in-line balanced pressure (ILBP) proportioner based on static and dynamic modeling. In order to automatically adjust the output foam concentrate pressure ($P_2$) in response to that of the supply water ($P_1$), the valve is designed as the poppet type featured in a four-chamber configuration. Unlike the conventional spring-loaded valve, the present design is novel in creating a cushion chamber on the back side of the poppet to provide the restoring force. Based on the working principles, mathematic equations for both static and dynamic characteristics of the valve are established, which have been simulated by the MATLAB code using the ode45 solver. The system exhibits a robust stability and performance, and it is insensitive to the variation in the fluid medium and initial condition. With reference to the geometrical parameters, the diameter of the diversion holes significantly affects the responsiveness of the valve, but the diameter of the valve stem nearly has no effect. Based on the numerical simulation and optimization, a physical prototype has been manufactured and tested in the laboratory, verifying its superior stability and performance. At the design condition ($q_e$ = 8 L/s, $P_1$ = 1.6 MPa), $P_2$ quickly reaches the equilibrium value $\widetilde{P_2}$ = 1.61 MPa which is very close to $P_1$, within a response time (or time lag) of merely about 0.01 s. The valve's start-up characteristics under different control pressures ($P_1$ = 1.2 and 1.6 MPa) and flow rates ($q_e$ = 4~10 L/s) have been examined systematically. The results show that the response time keeps constant at about 0.01 s regardless of the values of $P_1$ or $q_e$; the equilibrium outlet pressure $\widetilde{P_2}$ is dependent on $q_e$, but always approaches $P_1$ within the allowable limit. On the other hand, the equilibrium displacement of the poppet $\widetilde{X}$ increases monotonically with both $P_1$ and $q_e$, reaching a maximum of about 10 mm at $P_1$ = 1.6 MPa and $q_e$ = 12 L/s, which matches with the allowable limit to maximize the sensitivity and accuracy of the valve. Laboratory tests of the prototype verified that $P_2$ always follows $P_1$ without a noticeable time lag even when $P_1$ changes abruptly. The automatic pressure balancing valve developed in this study has been successfully put into production, forming the first series of the ILBP foam pump firefighting system in China.

**Author Contributions:** Conceptualization, Z.Y. and X.W.; methodology, Z.Y.; software: L.H.; validation, H.C.; formal analysis, Z.Y. and L.H.; writing—original draft preparation, Z.Y.; writing—review and editing, X.W.; supervision, X.W. All authors have read and agreed to the published version of the manuscript.

**Funding:** This research received no external funding.

**Conflicts of Interest:** The authors declare no conflict of interest.

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
