# Peer review of "Modeling and Design of the Automatic Pressure Regulating Valve in the Foam Firefighting System"

_processes, doi:10.3390/pr8121616_

Round 1

Reviewer 1 Report

The paper is clear and well written.

The topic of the paper is appropriate for the Journal.

The text of the paper was written correctly in terms of stylistically, punctuation and terminology.

The paper was correctly edited and graphically developed at a good level.

The literature was chosen correctly and fully used in the paper.

I do not see any shortcuts.

The paper deserves a positive assessment because it is current and interesting from both a cognitive and practical point of view.

Author Response

We wish to thank the Reviewer’s comments and recommendation.

Reviewer 2 Report

The article presents a new technical solution of a valve for automatic regulation of pressure in liquids. This solution has been reached in a modern scientific way – development of a dynamic nonlinear mathematical model, using this model to find the necessary solution using computer simulation and making a scale physical model using the simulation results. Based on these results, a real industrial solution was made.

My only doubt is whether the article corresponds to the topics of Processes.

Notes and questions:

  1. What is the step of integration (constant or variable) and the set accuracy in the computer simulation with Matlab?
  2. Line 131 - FIG. 4 instead of FIG. 3.
  3. There are some typos (eg line 138 - width instead of with)

Author Response

The authors wish to thank the Reviewer’s comments and recommendation. Below is the point-by-point response.

  1. During the numerical simulations, the step of integration was 0.0005 s and the root-mean-square (RMS) residual level of 10-4 was set as the convergence criteria. The above information has been added in the revised manuscript (at the end of the Section 2 – “Mathematical Model of Valve Motion”).
  2. The figure caption has been changed to “Figure 4” as suggested.
  3. The word “with” has been changed to “width” as suggested.

Reviewer 3 Report

The paper is written in the good way with the consideration of all necessary elements of the research paper. It contains experimental as well as numerical part, which is the main added value of the paper. The comprehensive description of the applied numerical set-up allows for it reconstruction using other than MATLAB numerical tools. The obtained numerical results shows good coherence with the experimental results. To sum up, paper is a good quality study on real engineering system and thus it is suitable for publication in Processes.

However, please improve quality of all plots.

Author Response

The authors wish to thank the Reviewer’s comments and recommendation.

As suggested by the Reviewer, all the plots (namely, Figure 5, 6, 7, 8, 9, 10, 11, 12a) have been re-drawn with a high resolution of 300 dpi.